# Evolutionary Perspective on Improving Mental Health

**Bjørn Grinde** 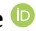

Division of Mental and Physical Health, Norwegian Institute of Public Health, N-0213 Oslo, Norway;
grinde10@hotmail.com

**Definition:** Mental issues are by many considered the main challenge for health authorities in industrialized nations. In this entry, I discuss an approach that may prove useful for ameliorating the situation and thereby improving quality of life. The analysis uses an understanding of the brain based on evolution and neurobiology, so consequently the ideas presented differ somewhat from traditional psychological thinking. Briefly, it appears likely that the problems with psychopathology are partly due to a lifestyle at odds with human nature. The key for finding preventive measures then is to identify the contributing mismatches. Based on the present perspective, therapeutic interventions can be construed as altering the brain by exercising functions that ought to be strengthened. By understanding brain plasticity, and the functions that are likely to need improvement in relation to mental health, we stand a better chance at devising interventions that work.

**Keywords:** brain exercise; environmental discords; evolution; mental health; neurobiology; therapy; well-being

## 1. Introduction

Mental disorders are arguably the worst health-related problem in terms of quality adjusted life years (QALY) and global economic burden [1]. According to estimates, close to half the population experiences a mental disorder during their lifetime, and 17–33% had a diagnosable condition in the last 12 months [2,3]. Anxiety and depression stand for a substantial share of the problems. The prevalence of diagnosable mental disorders may be just the tip of the iceberg; most likely, many more have reduced quality of life due to unnecessary worries and ruminations.

Anxiety and depression can be described as unwarranted activity in the fear and low mood functions of the brain, respectively. The activity of these functions presumably has a close-to-normal distribution in society (Figure 1). There is a somewhat arbitrary threshold for clinical diagnosis (the grey area in Figure 1). It seems unlikely that the present prevalence reflects the natural state for the human species, as one would expect these conditions to be selected against in ancestral populations [4]. A reasonable interpretation is that the situation is due to suboptimal aspects of the present environment.

I previously used an evolutionary perspective both to understand mental problems [4,5] and in categorizing psychopathology [6]. Other scientists have taken a similar approach [7–11]. I believe that this perspective offers relevant insight that may be missing in traditional psychological or psychiatric literature. In the present text, evolutionary biology is combined with current knowledge in neurobiology.

Mental health is typically related to two issues. One is a reduction in well-being, while the other involves difficulties with functioning in society. The two problems often appear together, but not always. A person struggling with anxiety can still function reasonably well, while people with Down syndrome and Williams syndrome require help, but appear to be happier than the average [12,13]. In affluent societies, we seem to do a reasonably good job at helping the disabled with practical aspects of life, which suggests that the main problem is to improve well-being.

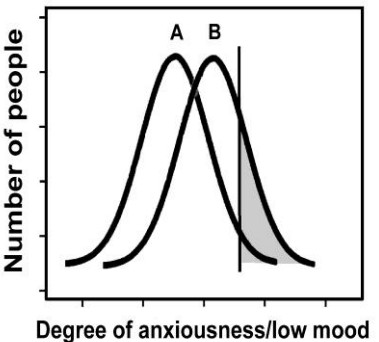

**Figure 1.** Theoretical distribution of fear and low mood. Above a more or less arbitrary cut-off (grey), the conditions are diagnosed as anxiety and depression, respectively. In an optimal environment (A), such as an ideal Stone Age situation, fewer people would be above the cut-off; while in a less healthy environment (B), which may include typical industrialized nations, more people suffer from these disorders.

Well-being and happiness are here used synonymously. The terms can be described as the mental part of quality of life. The concept covers contentment and subjective well-being, as well as both the hedonic (sensual pleasures) and eudemonic (human flourishing) aspects of gratification.

I shall focus on the well-being part of mental health for two reasons. One, as suggested above, is that this is where the shoe pinches in affluent societies; that is, most people with mental issues experience a suboptimal quality of life; and two, this is the issue that I believe has the greater potential for improvements. We can enhance well-being by creating an environment more suitable for mental development and by devising better therapeutic options. The problems related to functioning in society tend to be more deep-rooted, reflecting either brain deterioration (e.g., dementia) or unfavorable genes (e.g., schizophrenia [14] and Down syndrome). Although the practical needs of the sufferers can be taken care of, their handicaps are difficult to amend.

In the first part of the review, I describe a model of the brain that is suitable for the present purpose; the remaining text discusses approaches relevant to alleviating the problem in light of this model. Section 3 considers how society can move toward an environment better tuned to mental health. The fourth section is about therapeutic interventions; that is, how can we help people change in the direction of experiencing less pain and more happiness. The focus is on outlining an approach to mental health, rather than on offering an exhaustive list of options.

## 2. A Model of the Brain

### 2.1. The Key Question

The brain is an indispensable and delicate organ, and has consequently received special attention by the process of evolution. It is the only organ encapsulated by bone (the cranium), and it also has a blood-brain-barrier and a dedicated immune system. Although the brain has considerable plasticity in that the neural circuits are changing all the time, there is hardly any formation of novel neurons after birth. We start out with an abundance of brain tissue, as witnessed by the potential for recovery upon brain damage, but functionality is inevitably lost due to aging.

The brain cares for a variety of requirements associated with survival and procreation. A suitable approach to understanding this organ is to consider it as divided into numerous functional units, which can be referred to as *modules* [15–17]. I prefer the term applications, or *apps* [4]. Each app is engaged when required, somewhat like the apps of a phone, and deals with a need that arose during our evolutionary history. Brain apps are typically molded by experience, they involve dispersed neural circuits, and individual nerve cells may be active in several apps. Although we have considerable information as to which

parts of the brain are involved in various functions, the neural circuits have not been mapped in any detail.

There is one intriguing feature that distinguishes brains from any other part of the universe: they can create conscious experiences [18]. Human consciousness is based on input from a variety of apps, a list that includes those responsible for positive (pleasure) and negative (pain) feelings. Happiness, and thus mental health, depends on the balance between these two options; that is, the net activity of the apps [19]. The key question is how to coach the brain to provide pleasure and to avoid unwarranted pain. To have success in this endeavor, we need, for one, to understand what feelings are about, and two, knowledge on how we can influence conscious experiences.

*2.2. Why We Have Feelings*

The circuits responsible for the positive and negative components of feelings can be referred to as the *mood apps*. In order to understand these apps it is important to recognize why evolution added the capacity for feelings.

The primary purposes of nervous systems are to guide the organism to take advantage of anything that benefits the genes (opportunities) and to avoid anything with an adverse effect (dangers), as exemplified by obtaining food and avoiding predators, respectively. These two options form the basis of nervous systems, but evolution has devised a variety of schemes, or 'algorithms', to help decide what is the optimal behavioral choice in a given situation. Feelings represent one such algorithm. Presumably, feelings evolved some 300 million years ago in the early amniotes (the branch of the evolutionary tree consisting of reptiles, birds, and mammals); they offered the advantage of setting up a 'common currency' for evaluating options [20,21]. For example, when an animal approaches a waterhole, the fear of being eaten by a predator should be weighed against the pleasure of quenching the thirst.

In short, positive and negative feelings are there to lead us, respectively, toward what is good for the genes and away from anything harmful. The brain is designed to weigh the expected outcomes of actions based on the principle of maximizing the positive mood. In some situations, the resulting instigations have an immediate effect on behavior, but feelings are also important for classifying information in order to deal with future situations –the pleasure of success helps the organism remember that the strategy worked, whereas the pain of failure warrants a change in strategy.

The term *feeling* includes *sensations*, which reflect pleasant or unpleasant experiences associated with signals from external or internal sensing systems; and *emotions*, which encompass other forms of feelings, but typically concern social relations. All feelings have a positive–negative, or reward–punishment, component that is created by the mood apps. They also have a 'content' component, such as a sense of thirst or hunger, that is created by other apps. These other apps are normally responsible for activating the mood apps.

I previously argued that consciousness may have evolved as a consequence of the need to experience feelings [4]. It seems likely that other functions of the brain could theoretically be cared for in the absence of awareness, whereas feelings necessarily need to be felt.

*2.3. The Three Mood Apps*

The positive and negative element of feelings can be described as being based on three different mood apps [19,22,23]. Negative feelings rely on a single *pain* (or punishment) *app*. Whether it is a question of social punishment (such as shame) or a physical injury (braking a leg), the same neural circuits deliver the pain part of the experience. The reward system, however, is divided into a *seeking* (or motivating) *app,* which is meant to stimulate the individual to seek opportunities; and a *liking* (or consuming) *app*, which make sure the opportunities are utilized once available. The two apps are exemplified by respectively following the smell of a bakery and eating a cake; they can be referred to as a combined *pleasure* (or reward) *app*.

The above classification reflects not only present knowledge as to the neural circuits responsible, but also evolutionary theory [4,23]. An animal must seek opportunities before it can consume what it finds, which explains why the two needs are organized as separate apps. Dopamine and serotonin are the core neurotransmitters for the seeking app, while endorphins serve a similar role in the liking app.

There is substantial evidence supporting the notion that all forms of pleasures and pains converge on the above three apps. For example, experiencing envy of another person's success activates pain-related circuitry, whereas experiencing delight at someone else's misfortune activates reward-related neural circuits [24,25]. Similarly, feeling excluded activates pain-related neural regions [26], while positive social experiences, such as being treated fairly and cooperating with others, offer rewards similar to those obtained from desirable food [27–29].

Several brain apps can activate the mood apps, causing various experiences to be considered either pleasant or unpleasant. Fear, for instance, normally activates pain, but can also activate pleasure, as in the case of the adrenalin kick of a climber. The distinction between mood apps and activating apps is based on the present neurobiological and evolutionary perspective. This dichotomy is not typical for psychological textbooks, and is, for convenience, not strictly adhered to here. For example, when discussing how to reduce the conscious contribution of the fear app, as in treating anxiety, the obvious aim is to avoid negative feelings—not the joy of climbing.

'Dimming switches' rather than simple 'on-off' buttons control the mood apps. Mood consequently varies on a scale that ranges from very unpleasant to highly pleasant. The switches for the mood apps belong (primarily) to the nonconscious part of the brain—for the obvious reason that they are meant to control you, rather than you controlling them.

The mood circuitry can be activated directly from a sensory experience, such as tasting sweet food or burning a finger; however, cognitive deliberations may intervene to the effect of either subduing or enhancing the pleasure or pain. Cognition can also instigate feelings in the absence of external stimulation, as when daydreaming. Minor alterations in a situation or a line of thought—whether due to conscious input, subconscious brain activity, or external factors—can change the net effect abruptly, as when the positive feeling of the climber ends if he/she suddenly slips and falls.

It is important to note that negative feelings are pathological only if inappropriate or in excess. Proper forms of agony serve a function, while the inability to feel fear [30] or physical pain [31] is a severe handicap. Any activity in the pain circuitry will decrease happiness temporarily, but the lack of appropriate activity can cause a much more severe reduction in the long run. Strong positive feelings are rarely a problem; however, one may argue that the manic phase of bipolar disorder [10], and perhaps certain forms of recklessness, reflect an undesirable hyperactivity of reward circuits.

The process of generating awareness can be depicted as having the various apps sending out bubbles of information from their position below the surface of consciousness (Figure 2). The bubbles that reach the surface (the process takes some 300 ms) fuse to form a moment of experience [32]. The beholder of the brain experiences the combination, but is generally unable to separate the different contributions. For example, you do not recognize that envy implies activity in the same pain app as when feeling excluded.

### 2.4. Default Contentment

Most of our daily activities have limited relevance for the level of happiness. Most people do not experience life as a stream of either good or bad events, but as a relatively steady state. Mood typically moves slightly up or down, as when working on an interesting task or being worried, respectively. More rarely, particular incidents may cause a surge of pleasure or pain. In other words, the mood apps do not normally dominate the mind, but that does not mean they are inactive. It seems more appropriate to envision a tonus of mood caused by a balance of positive and negative activity. The steady state tonus presumably reflects what has been referred to as a setpoint of happiness [33]. It is relatively

easy to find stimuli that send happiness temporarily beyond the setpoint, while it is more difficult, but not impossible, to enhance the setpoint itself.

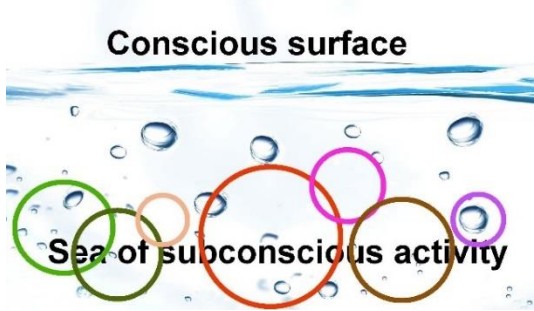

**Figure 2.** A model of consciousness. The brain can be portrayed as a sea of activity, where the surface represents the conscious. The functions, or apps (depicted as circles), are embedded in the subconscious (or nonconscious) sea. Some are more important as to conscious content (larger circles) than others. From there, they send bubbles toward the surface. The bubbles that reach their destination fuse to form a moment of conscious experience.

The dichotomy of hedonism and eudemonia is an important issue when discussing happiness [34,35]. It is relevant to note that although the present model suggests that the activity of reward circuits defines well-being, it does not promote hedonism. Typical hedonic stimuli are most likely of limited importance for the mood value when integrated over a lifetime. For one, hedonic pleasures tend to be of short duration, and may at some point cause harm [36], and two, hedonic stimuli are not required to activate the reward circuitry, as eudemonic options serve the same purpose. For example, engaging in meaningful activities is associated with rewards, presumably because evolution promotes any pursuit considered useful.

Humans, as well as other mammals, appear to be designed to be in a good mood as long as there is no cause for negative feelings. This is presumably because the interest of the genes is best cared for when residing in an individual with a positive attitude to life, as this state is more conducive for survival. This idea, which has been referred to as a *default state of contentment*, is discussed in more detail elsewhere [19]. Retaining this state of mind is probably the crucial factor in mental health. The problem is that so many people suffer from needless negative feelings, particularly those related to fear, low mood, and physical pain. To enter the default state of contentment, and thus enhance the setpoint of happiness, negative feelings have to be turned down.

### 2.5. Implications for Mental Health

The human brain is perhaps the most plastic or malleable of any brain due to the importance of postnatal development and learning. The plasticity is primarily a question of changing the connectome (the wiring that connects neurons), and we have considerable knowledge as to how this is achieved [37–39]. At least three different mechanisms are at work: one, the formation and pruning of synapses; two, the changing of synaptic properties to make them more or less efficient at transmitting signals; and three, increasing the amount of myelination-wrap on the axons to speed up signals. Neurotrophins are key actors in the transformation of neural circuits. They are produced as a result of neural activity, which means that the neural circuits being engaged is the focus of change.

The brain can be rendered more malleable to modifications by, for example, chemicals such as D-cycloserine [40], or by stimulation of the vagus nerve [41]. This point is relevant because mental therapy generally implies an attempt to alter neural circuits. The inherent plasticity suggests that the brain theoretically can be fashioned somewhat to the desire of the beholder. The process of change resembles physical exercise in that both muscles and neural tissue tend to strengthen and expand upon use and degenerate in the absence of

use [42,43]. Brain circuits can be boosted upon repeated activation, to the extent that it is possible to measure an increase in volume of the brain area [44]. If the exercised function impacts on conscious experiences, that impact is expected to increase. For example, by regularly engaging compassion, one becomes more compassionate [45,46].

As opposed to muscles, there are neural circuits you do not want to strengthen, in particular those that turn on pain. This point is illustrated by the observation that people who suffer from chronic pain tend to become progressively more sensitive to any kind of ache [47]. By focusing attention on the original pain, the patient apparently enters a negative feedback cycle that enhances the responsible apps.

Negative feelings typically involve defense functions; fear and physical pain are there to prevent you from being killed or damaging your body. This is more important for the genes than pursuing pleasures; consequently, negative feelings tend to be more easily triggered, which means they are more likely to become exercised. For example, by regularly experiencing fear, you increase the risk of developing anxiety problems [48].

The evolutionary rationale for low mood is not as obvious as in the case of fear and physical pain. Most likely, the pain associated with a low mood is meant to warn you that things are not quite as they ought to be. People feel upset when something goes wrong and happy when they succeed. In the Stone Age, it could be a question of missing a prey, today perhaps flunking an exam. The discomfort instigates a desire to devise new strategies for dealing with the task.

Negative feelings tend to dominate the mind. Their importance implies that while activated, it is difficult to feel pleasure. Those who suffer from chronic pain, anxiety, or depression often experience apathy and anhedonia [49]. Unfortunately, the various forms of pain seem to be less subject to conscious influence. The brain is designed to seek rewards, and you are expected to make personal choices in this regard, while the task of reacting to potential dangers is to a larger extent delegated the nonconscious brain. You decide to go out and look for food, but not to seek a hungry lion.

### 2.6. Variations in Impact

As opposed to muscular exercise, it seems that the impact of mental events varies considerably depending on how the situation is conceived. The brain is designed to take special note of experiences that engage feelings. If an event causes a strong activation of the mood apps, it implies that what happened has considerable importance for fitness. The event ought to be heeded, which means the accompanying feelings are more likely to affect future conscious experiences. Negative events are particularly relevant, due to their consequences for survival, which help explain the problematic condition referred to as post-traumatic stress disorder.

A positive experience can also have a substantial impact, as exemplified by how some people react to the therapeutic use of psychedelics [50]. Moreover, even traumatic episodes can turn into something positive if the individual manages to deal with the situation in a constructive way, an effect that has been referred to as post-traumatic growth [51].

A moment of experience consists of contributions from a variety of apps that presumably compete for conscious access [32,52]. In the present model this means that only some bubbles manage to reach the surface (Figure 2). If one app, such as fear, is very active, other apps are likely to be suppressed.

### 2.7. Good and Bad Habits

Habits reflect a form of brain exercise where neural circuits change with limited effort. Habits are based on the fact that the feelings associated with an experience shape future preferences. If something causes pleasure, we want more of the same. The neural circuits responsible for the relevant behavioral pattern are therefore strengthened in order to promote the action that gave the reward. Most people are under the spell of both good and bad habits.

If a bad habit is strong enough, we refer to it as an addiction. Heroin hits the pleasure app with considerable force, therefore, the brain naturally assumes you have hit the jackpot; that is, you have found something worth a lot of effort to obtain.

Habits can violate the principle that your brain is set to optimize positive feelings. A heroin addict is unable to avoid the craving and the injection, even if the act causes more misery (in the form of remorse) than pleasure. The behavior has become deeply engrained in the nervous system. The conscious brain is allowed an attempt to modulate the action, as by saying it is best to avoid the shot, but this input is not necessarily sufficient to block either the urge or action. The situation exemplifies the power that the nonconscious brain can have on conscious decisions.

Obsessive-compulsive disorder (OCD) is another example of habit-formation going awry. We know a bit about the neurobiology involved in compulsive behavior; the striatum and parts of the cortex are involved [53]. OCD is associated with serotonin disturbances in these parts of the brain, a neurotransmitter known to play a role in the seeking app [54]. The brain treats the compulsive behavior as a unit to be activated when the right cue appears. The person receives drops of pleasure upon repeating the behavior, but the drops can be too small to be recognized, and they can be overshadowed by a negative assessment of the compulsion.

Humans are probably designed to rely more on cognitive resources compared to other mammals, and therefore less on innate, pre-programmed behaviors. However, decisions based on cognition tend to be slow and costly in terms of brain resources, so habits ease the process of deciding what to do. Perhaps our cognitive capacity has made us more prone to habit formation, which again may explain why we so easily are caught up in undesirable routines, such as addictions and compulsions.

## 3. Improving the Environment

### 3.1. Mismatches and Discords

The mental health and happiness of an individual depends on three factors: for one, roughly half of the variation in well-being, as well as the disposition for disorders such as anxiety and depression, is due to genetic inheritance [55,56]; two, the environment counts for the remaining disposition; but three, it is possible to improve the impact of the environment by therapeutic interventions.

For all practical purposes, there is nothing we can do about the genes, so I shall focus on environment and interventions. As the brain is most malleable in the early years, childhood experiences can have a major impact on how adults perform [57]. There are sensitive periods for the maturation of neural circuits during infancy, and these periods are relevant not only for sensory or motor circuits, but also those involved in the development of feelings [58,59]. The high prevalence and early onset of psychopathology suggest that the present infant environment is not optimal.

One way of describing the problem is to say that an environment at odds with what is natural (or preferable) for the human species will push those with vulnerable genes toward mental issues. The concept *environment of evolutionary adaptedness* (EEA) has been coined to suggest the type of environment in which we are designed to flourish [60]. It can be thought of as the optimal Stone Age environment, as the modern situation is typically worse in some respects but better in many others. That is, most differences, or *mismatches* [61], between the present setting and the EEA are presumably either neutral or beneficiary. We sleep better on soft mattresses rather than on damp soil, and we live longer and healthier thanks to antibiotics. Mismatches that contribute to pathology can be referred to as *discords* [62].

To initiate preventive measures, we first need to pinpoint the discords responsible for mental problems. While it is reasonably simple to suggest mismatches, it requires dedicated research to identify, and estimate the impact of, actual discords. Near-sightedness offers an illustrative example. The contrast in the prevalence of near-sightedness when comparing people living in cities (up to 80%) compared to rural areas (typically 1%) [63] suggests the involvement of discords. The leading candidates, in the form of likely mismatches, were

for one, focusing on a close and fixed distance (as when reading); and two, not having the natural diurnal cycle of light and darkness (lamps being on at night). However, research suggests that the main discord is the lack of time infants spend outdoors, as the eyes require a certain amount of UV radiation in order to develop properly [64].

The consequences of discords on human health have been referred to as the diseases of modernity; the high prevalence of anxiety, depression, and chronic pain suggests that these conditions should be included [11,62,65]. The prevalence of nearsightedness can be reduced by exposing infants to UV radiation; hopefully, we will find ways to lessen the mental burden as well.

*3.2. Likely Discords*

It is beyond the scope of the present text to set up a comprehensive list of possible discords affecting mental health, but I shall offer some examples to suggest a way of thinking.

The high prevalence of anxiety related disorders indicates that the normal way of caring for children is not optimal [66]. Infants are unable to comprehend the concept of predators or burglars, so for them proximity to a caretaker is the key to feel safe. Preferably there should be skin-to-skin contact, as pleasant touch mitigates stress and the feeling of isolation in both infants and adults [67]. The present way of handling infants typically involves reduced parental proximity; for example, a separate crib and strollers instead of carrying. It is well known that the stress of infant separation can cause susceptibility to later anxiety disorders [48]. It seems likely that milder forms of similar stress, such as insisting that the infant sleeps alone, also increase the vulnerability by exercising the fear function.

The Western style of parenting stands in contrast to what is typical for indigenous populations. The biologist Jared Diamond has, for example, described tribal people of New Guinea [68]. He points out that parents have frequent skin contact with infants, mothers breastfeed until the child is relatively old, adults are within reach around the clock, and they respond quickly if the child cries. It should be possible to promote similar behavior in modern societies, and if we do, there is reasonable hope that the mental burden will diminish.

Perhaps the most difficult discord to deal with concerns social life. Social relations are of prime importance for well-being [69–71]. Humans became social late in our evolutionary history, and at that stage behavioral instigation was presumably cared for primarily by positive and negative feelings, rather than more instinctive or fixed forms of control [72]. As a consequence, interpersonal connections appear to stand for a considerable part of both pleasures and pains.

In the Stone Age, most people spent their entire life in a small group. The tribal setting offered safety, whereas being alone was a perilous situation. As each person depended on the others, they presumably managed to maintain good relations. Reinstating the social setting of the Stone Age is difficult in modern societies, as industrialization depends on gathering hordes of strangers in cities. Many people are troubled by loneliness because they lack a proper social network. It may be possible to ameliorate the situation by setting up 'villages' within cities. Individuals living in small-scale, intentional communities appear to be happier than the surrounding population [73].

As to the social environment of children, the Stone Age tribes normally offered a number of peers of a similar age to play with. Today, most children grow up in small family units. The problem is illustrated by a study suggesting that a larger number of older siblings protects against later development of emotional disorders [74]. We send children to kindergartens, but these only provide company for a limited part of the day, and the peers are more likely to change.

A low mood is called for when expectations are not met. The low mood app seems to be particularly responsive to social relations; the unpleasant sensations of loneliness or being excluded are meant as an instigation to improve your network. Consequently, lacking proper social connections contributes to depression [75]. Infants who miss parental

proximity are expected to have both their fear and low mood apps activated, which may help explain why anxiety and depression so often appear together [2].

### 3.3. Lifestyle

The discords suggested in the previous section are related to interpersonal connections. Social life is likely of particular importance, but mental health is also entwined with somatic health, which means that factors impacting on health in general are likely to influence the mind. Healthy nutrition [76] and appropriate physical exercise [77] matter, and both are most likely not optimally cared for today. One effect of the latter is to boost the mood-enhancing neurotransmitter dopamine [78].

Chronic pain is not necessarily considered a mental health problem, but it seems appropriate to include it here. For one, the same pain app is activated whether it is a question of physical or mental suffering [23]; and two, chronic pain has a considerable impact on well-being by affecting some 20% of the population [79]. Lifestyle factors such as inactivity, obesity, stress, and a lack of sleep contribute to the high prevalence of chronic pain [80].

In the case of anxiety and depression, the pain app is turned on by activity initiated within the brain, whereas physical pain is different in that it (normally) originates in peripheral sensory cells referred to as nociceptors. Two common causes of chronic pain are inflammation (such as in arthritis) and neuropathy (which stems from damage to nociceptors and surrounding tissue), but chronic pain can also be sustained by the central nervous system in the absence of peripheral input [81].

Addiction is another major problem in industrialized nations. Beyond the fact that a larger variety of addictive narcotics are more easily available today, a suboptimal environment most likely increases the chance of addiction [82]. In the absence of default contentment, the pleasures associated with hedonic stimuli, whether in the form of drugs or chocolate, appear stronger. That is, the gratification is less distinct for those who already are in a good mood. The stronger the pleasure, the stronger is the desire for more, which leads to either bad habits or addiction.

Problematic habits belong to the diseases of modernity. In the Stone Age, there were not that many temptations, whereas today all sorts of stimulants are easily available. Restraint has become an important virtue, and the ability to resist temptations a necessity. In other words, the key to success in an affluent society is to establish good habits and avoid bad ones.

## 4. Therapeutic Intervention

### 4.1. How to Exercise the Brain

Ideally, the environment should leave everyone with a healthy and happy mind. Unfortunately, mental problems most likely troubled people in the Stone Age as well, and will continue to do so in the foreseeable future. If we can create an environment incorporating the best of both the EEA and industrialization, the prevalence should go down; however, it seems unlikely that we will obliterate the need for additional measures to alleviate suffering. The human mind is too vulnerable.

The challenge of therapy is to generate desired alterations in adult brains. In most cases, the preferred change is to coach consciousness to deliver less negative and more positive feelings. Typically, it is a question of turning off excessive fear, low mood, and physical pain; in other words, to cure anxiety, depression, and chronic pain, respectively. The interventions can be described as individually tailored exercise programs for the brain.

The brain is designed to be formed by both external affairs and activity initiated from within. The nonconscious brain feeds information to the conscious, but there are channels facilitating impact in the opposite direction. In other words, it is within our power to 'hack into' nonconscious processes. Consequently, it is theoretically possible to exercise most, if not all, relevant circuits. The question is, how to exert the desired impact?

The pupils of the eyes offer an illustrative example [83]. The muscles in the iris regulate their size by contracting upon increasing light and dilating when it gets darker. A nonconscious reflex controls the process, but you can hack into the neural circuits responsible simply by envisioning something dark or light. Therapy presumably depends on finding similar interventions.

An important difference between mental and physical exercise is that muscles are more easily accessible. The big challenge, in the case of the brain, is to devise suitable training regimes. The task is easier if the relevant neural circuits can be engaged at will, and progress is easy to measure. Individuals who suffer from a stroke demonstrate the potential for training brain functions. If, for example, the stroke has destroyed the neural circuits responsible for controlling movement of a leg, the patient can recover the ability to walk. In this case, you know exactly what you want the brain to achieve, and improvements are observed in the form of movement. The routine is relatively easy to set up, start by using crutches and gradually training the leg to do the job, but the task requires repeated attempts to activate relevant muscles many times a day over several months [84].

The apps responsible for feelings are most likely malleable, but it is more difficult to devise suitable routines for exercise because feelings, as opposed to leg muscles, are primarily under the control of the nonconscious brain. On the other hand, progress in the case of a stroke likely requires the establishment of new neural circuits in healthy parts of the brain, while strengthening the capacity to turn off fear should only need modification of the existing circuits.

If you want to improve your biceps, you lift weights. As in the case of patients recovering from a stroke, the exercise needs to be repeated many times over a long period. However, a single experience sometimes has a profound effect on the brain. Post-traumatic stress syndrome is a well-known example. Another difference between muscles and brain is that the latter can be exercised even while asleep [85]; what matters is the activation of relevant circuits, not whether you are aware of what is happening.

Anything that impacts on the brain implies an element of exercise, in the same way that the daily use of your body exercises muscles. The fact that the adult brain is considerably different from that of a baby, in terms of both emotions and knowhow, testifies to the considerable changes that occur even in the absence of explicit exercise. In fact, most of the molding of the brain is probably cared for in the absence of any deliberate attempts, as exemplified by the way infants learn language.

It is easy to enhance specific cognitive skills [86], but the training regimes do not offer any general improvement in cognition [87]. The chess enthusiast boosts his performance in chess, but not in bridge. This point reflects the general principle that only the circuits activated are strengthened.

Presumably, there are neural circuits whose function is to either switch on or switch off various apps (Figure 3). In nature, a frightening situation will generally resolve itself within a short time, and to avoid the accompanying fear obstructing other activities, the feeling is turned off. Inappropriate development of the fear function probably results when the 'on-button', but not the 'off-button', is regularly engaged. In short, worries that persist without resolution may dispose for later anxiety problems; the capacity to keep fear on is strengthened, but not the capacity to turn it off. If parents put their infant to bed in a separate room, the child may feel the absence of caretakers as a treacherous situation, and resolution does not come until the next morning. A quarrel with the neighbor or at work may turn into a similar prolonged and stressful situation.

Focused exercise is needed to improve the capacity to escape worries. A psychologist may help by suggesting strategies for training, and by offering encouragement, but as in the case of the patient suffering from a stroke, the exercise needs to be cared for by the individual. We live in a world where there is a tendency to assume that doctors can cure any disease. The assumption usually works in the case of an infection or a hernia, but not with mental disorders. People need to be informed that a personal commitment is required to alleviate negative feelings.

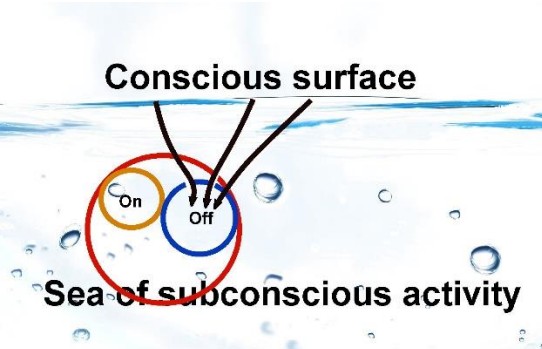

**Figure 3.** Exercising relevant apps. Although the apps that control feelings belong to the nonconscious brain, there are ways to use conscious processes to impact on their activity, as indicated by the arrows. The idea may be to strengthen the capacity to turn off negative feelings, such as those generated by the fear app (red circle).

I divided the typical strategies used for intervention into four categories: medication, traditional psychological therapy, coaching or self-help, and advanced tools for exercise. The categories will be discussed in the following four sections. The point is not to offer detailed suggestions for practice, but to offer a theoretical framework.

*4.2. Medication*

Medication with psychoactive substances is a key component of present treatment. Although standard medicine only helps a fraction of patients, the pills have alleviated considerable suffering [88]. A substantial component of the effect is likely placebo, illustrating a substantial cognitive impact on all sorts of pain, but that component is valuable and thus worth an effort to exploit [89].

Most of the substances used affect synaptic transmission by binding to the proteins involved in handling neurotransmitters. A main problem is that the same neurotransmitters are involved in a wide array of functions all across the brain. The problem is ameliorated by the fact that key neurotransmitters, including dopamine and serotonin, interact with a considerable number of different proteins. The various proteins are, to a larger extent, associated with particular functions, allowing for some specificity in drug design. However, the present arsenal of medicine is hardly ideal.

One problem is that although the drugs may mitigate symptoms by having an impact on the relevant brain circuits, they are unlikely to cause the desired changes in these circuits. The most important effect is arguably to give the patient the relief and energy required to exercise the mind. Thus, medication should ideally be used in conjunction with therapy or other forms of training. One might add that although subduing the activity in apps such as fear and low mood does not count as exercising the capacity to turn them off, it may prevent unnecessary activation that would further strengthen them.

Psychedelic substances, such as ketamine and psilocybin, offer a novel form of medication [90]. These substances have a reasonable track record in alleviating major depressions and addictions, but we have limited knowledge as to how they work. An interesting observation is that while other medications generally require a presence in the brain, psychedelics seem to instigate changes that last after the substance has left the body. The effect may be related to the profoundness and power of the experience. The cataclysmic event may help the patient restructure the brain, somewhat as in the case of post-traumatic growth. There is evidence that at least ketamine increases the plasticity of the brain [91], which may also help explain the lingering effect.

To summarize, we do have medications that can ease the burden of at least a substantial fraction of those who suffer from mental disorders, but alone they rarely function as a cure. As a rule of thumb, medication is best used in conjunction with other forms of intervention.

This conclusion is in line with the idea that desired changes in the brain usually require a committed exercise effort by the individual.

### 4.3. Psychological Therapies

Psychology and psychiatry are here lumped together.

Cognitive (or behavioral) therapy in the treatment of phobic anxiety is arguably the most successful form of therapy [92]. The patient is required to enter a situation that triggers fear, the point being that the situation offers an opportunity to exercise the capacity to turn it off. For example, if the problem is an irrational fear of spiders, one starts with a picture of a small and innocent example. As conscious deliberations impact on the issue of whether a situation actually poses a threat, the patient can learn to turn off the fright that arises upon seeing the picture. One moves on to stronger images and eventually learn to subdue the reaction to live spiders. This training is in line with the idea that while it is difficult to block the initial triggering of fear, it is possible to learn how to stop the response before it has time to unfold in the brain. The success of this form of therapy probably rests with the ease of setting up an exercise routine.

Cognitive therapy can be used to improve other mental problems as well, including depression [93]. The therapy presumably works by first identifying the negative, self-destructive thought patterns, and then finding a way to change the mind towards a more positive attitude. In the present terminology, this implies exercising switches that help turn the low mood off. There are numerous methods by which this can be achieved in a therapeutic setting; the idea is typically to talk with the patient, and thus summon thoughts that are likely to contribute to the pursuit.

Physical pain offers another example. Inappropriate pain is a considerable challenge when it comes to developing brain exercise strategies, but there are some options [94]. One is to avoid focusing on the pain, for example, by telling yourself not to care. A related option is to use distractions to lead the mind away from the pain, for example, by getting involved in a film. In both cases, the patient improves the capacity to evade the pain, which means turning off the conscious impact. The strategy is based on the fact that the brain is designed to curb pain if there are more pressing issues in need of attention. For example, if a soldier is wounded in a battle, it makes sense to postpone the pain reaction until he has reached safety. Although the decision to postpone pain receives a major input from the nonconscious brain, it is possible to 'fool' the brain to consider other things as more important.

Fakirs, as well as specially trained Buddhist monks, have proven that it is feasible to train the brain to not be bothered by incoming pain signals. In fact, the corresponding reduction of activity in pain related circuits can be measured [95]. Physical pain is perhaps the form of pain that is furthest from cognitive control, thus their success suggests that all negative feelings can be eased by related strategies. It should, however, be noted that the Buddhist monks spend years training.

### 4.4. Coaching and Self-Help

As pointed out above, improvements generally require a solid commitment in the form of regular practice. The result is likely to improve with expert guidance, but as you need to do the exercises yourself, it is possible to go on without help. Most people may prefer someone to aid them, but an informally trained mentor or coach can suffice.

Aside from advice on strategies and inspiration to actually perform the exercise, there is an additional advantage of seeking help. We belong to a highly social species; apparently, we have an innate tendency to assume that people we associate with are there for us. This means that by talking about your problems, you feel that part of the load is lifted off your shoulders and onto the listener. Your coach offers this form of social support, but a friend may serve the purpose equally well.

Most people actually experience a burden when talking to a troubled person, but the listener is normally compensated by harvesting the pleasure of compassion. Moreover, the

listener does not really need to take on the load; those who practice therapy as a profession learn to appear empathic without necessarily letting the patient's problem weigh them down. If in lack of a person to talk with, it helps to consider a diary as a confident; or, for those religiously inclined, to get in touch with God.

One way of using the advantage of interpersonal connections is to join a support group. The group may consist of peers—individuals with a background in similar problems [96]—but any form of congregation can do. A group offers more people to share your troubles with, and an increase in the positive effect on mood that comes with socializing.

While most psychologists or psychiatrists tend to focus on negative feelings, personal coaches more often focus on how to enhance positive mood. Moving towards more happiness is the ultimate objective; unfortunately, negative feelings tend to take precedence, due to their importance for survival. Consequently, exercising their off-switches may be required in order to progress. On the other hand, it seems easier to activate the switches that turn on pleasure rather than those that turn off negative feelings, as the latter are more in the hands of the nonconscious brain. Doing, or thinking about, something pleasurable, while at the same time trying to engage in the enjoyment, is a reasonable recipe. Simply forcing the mouth to form a smile may be sufficient, as the smile muscles apparently activate the pleasure app [97].

Meditative techniques have a long tradition in alleviating suffering and improving mood [98]. Tibetan Buddhists claim that their practice is capable of molding a pleasure app of sufficient strength to allow for a positive mood regardless of the situation [99,100]. A key element of meditation is to disconnect consciousness from the plethora of external and internal commotions; that is, to diminish the input from thoughts and sense organs (Figure 4). The cessation of these disturbances leads you into a meditative state, a condition described by terms such as peace and harmony.

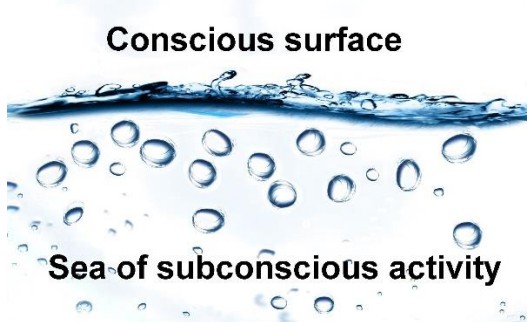

**Figure 4.** Meditation. The nonconscious brain normally sends a lot of bubbles to fuse at the conscious surface. Meditative techniques reduce this flow. By calming the mind, it becomes more accessible to exercise.

Meditative techniques offer three possible exercise strategies (see [4] for a more detailed discussion). One is the traditional idea of moving the mind into a relaxed state. By reducing the quantity of thoughts and emotions, you inevitably tune down undesirable activity. The second strategy applies particularly to the school of meditation referred to as mindfulness. The practitioners are relaxed, but at the same time they exercise the capacity to stay focused. The technique apparently helps people focus on things other than negative thoughts, even when not in a meditative state [101].

The third option combines the previous two. The meditative state provides a state of mind that is suitable for exercises aimed at specific apps. In short, by clearing the mind from distractions, it is easier to engage the desired neural circuits. Mindfulness helps you to obtain the focus required to do so. Sentences such as 'I feel fine' and 'nothing to worry about' can be used as mantra to activate the desired circuits, including those responsible for turning off anxiety. Forming sentences, silently or aloud, is sufficient to impact on the mind, and thus exercise the relevant circuits, particularly if you manage to conjure up the

sentiment suggested by the sentence [102]. Therapeutic approaches, such as mindfulness-based cognitive therapy and mindfulness-based stress reduction, exemplify the use of this strategy to turn off negative feelings [103].

Tibetan Buddhist monks have a tradition for using this third option. Their meditation is often directed towards compassion. By using short sentences such as 'I wish you to be happy', the practitioner engages this app. There are at least two benefits to harvest from strengthening the capacity for empathy: for one, you activate rewards associated with socializing; and two, by becoming a nicer person, you are likely to improve your social connections. In a way, the exercise may be considered selfish, but the feeling needs to be genuine in order to work, which means that the practitioner also becomes more caring.

As a general rule, engaging in anything enjoyable will exercise the pleasure app. Music, for example, has the potential for improving the mind [104,105]. Another option is simply to focus on positive thoughts, for example, by taking time every evening to write down good things that happened during the day [106].

*4.5. Exercise Tools*

The most difficult part of exercising the mind is finding a suitable workout routine, which means a way to activate the neural circuits one wants to strengthen. The concept of neurofeedback suggests a solution [107]. The idea is to measure the activity in the relevant circuit and translate this to a suitable feedback signal, such as the intensity of a sound. The practitioner then focuses on how to increase the intensity by trial and error. The situation can be compared to the way in which infants learn to control their hands; they just try out options, without any idea of how to obtain the desired result. For the child, the motivating factor may be the ability to build a tower of bricks; for the adult, to escape anxiety or depression.

There are two main obstacles with this strategy. For one, it is difficult to measure activity in the neural circuits one wishes to improve. Although we do have knowledge relevant for gauging, for example, fear [108], the measures are not ideal, and are not adapted to individual issues. Two, it may be difficult to find a way to enhance the signal, even with relevant feedback information.

The easiest way to measure brain activity is by electroencephalography (EEG). EEG-neurofeedback has been used for several decades and for a variety of purposes, including both the treatment of mental conditions and cognitive enhancement [109]. Manipulating the EEG levels is relatively easy, but so far, the strategy has not lived up to the initial enthusiasm in regard to curing mental disorders. EEG only reflects activity in the outer part of the cortex, and one cannot expect this to be the seat for functions such as fear or low mood. Functional magnetic resonance imaging (fMRI) is probably more suitable as a measure, but inconvenient to set up. Experiments with neurofeedback based on fMRI have had some success in reducing depression and chronic pain [110,111].

It is possible to directly activate parts of the brain with electrodes or with transcranial magnetic stimulation. The approach implies that the patient does not need to find a way to exercise the brain, but can let machines do the job. Experimental use suggests that magnetic stimulation can alleviate certain forms of pain [112].

## 5. Conclusions and Future Research

Improving mental health is likely to remain one of the biggest health challenges for years to come. The challenge has at least four components: one, it is important to define the environmental factors that push the brain in the wrong direction; two, we need to improve our understanding of the relevant neural circuits; three, we need ways to engage these circuits in order to exercise them; and four, it would help if we had better pharmaceuticals, preferably aimed not only at alleviating symptoms but also improving the condition. I believe the present perspective offers insight that is supplementary to traditional psychology when facing this challenge.

Finding relevant discords is particularly important, but unfortunately difficult. Ideally, one should test candidates by randomly assigning subjects to different environments, but in the case of humans, this option has obvious limitations. Instead, research has to rely on correlations and studies on animals.

There is considerable comorbidity in that patients often are diagnosed with more than one condition. Two common examples are anxiety/depression and schizophrenia/bipolar. In the former case, the explanation may be partly due to both conditions resulting from the same experiences; in the latter that the actual problem is the borderline between the two diagnostic outcomes. Somewhat surprisingly, all pairs of mental disorders, including those one might expect to be distinct (such as schizophrenia and autism), tend to appear together more often than chance [113]. This observation suggests that there are either genetic or environmental factors that cause a general mental imbalance [114]. The idea is supported by evidence indicating that an unspecific therapeutic approach can serve as well as more disease-oriented protocols [115]. An important challenge for future research is to pinpoint the relevant factors.

Over the coming decades, we will hopefully see substantial progress as to neurofeedback forms of exercise. For that to happen, research needs to focus on better methods of measuring activity in the circuits one wishes to enhance. Whatever you feel or think, there is necessarily a neurological correlate in the brain. As proof of principle for this statement, one has been able to instruct a computer to distinguish between different thoughts, such as a house versus a car, based on brain data [116]. It should be possible to obtain more accurate measures for the circuits responsible for regulating mood.

There is also hope for finding better medication. Psychedelics seem to have potential; the list includes not only ketamine and psilocybin, but also cannabis and the empathogen MDMA [90]. More research is needed to find out how to best exploit this option, including when to use which substance and what sort of additional therapy to include.

It is possible that evolution has installed particularly strong feelings in humans. Our cognitive capacity left the genes vulnerable to the whims of behavior, as evolution may have used stronger incitation by the pleasure and pain apps to reduce the chance of undesirable consequences. This assumption is supported by the observation that endorphins, which are key neurotransmitters in relation to pleasure and pain, are expressed at higher levels in human brains compared with apes [117]. In other words, we may have the potential for being both the happiest and the most miserable species. Industrialized countries struggle to handle the human predicament in an optimal way.

Present top runners would most likely outpace those of the Stone Age due to our knowledge regarding physical exercise. Science ought to focus on bringing similar expertise to the challenge of improving mental health. The aim should be to design a society where people get top scores on happiness. Based on the prevalence of mental problems, there seems to be a considerable way to go.

**Funding:** This research received no external funding.

**Institutional Review Board Statement:** Not applicable.

**Informed Consent Statement:** Not applicable.

**Data Availability Statement:** Not applicable.

**Conflicts of Interest:** The author declares no conflict of interest.

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
