# Peer review of "Evolutionary Perspective on Improving Mental Health"

_encyclopedia, doi:10.3390/encyclopedia2030099_

Round 1

Reviewer 1 Report

1.     The subject of this paper is relevant today as mental health is a tough and timely society concern. The author’s comment on the known prevalence of mental disorders may be just the tip of the iceberg is particularly apt as mass shootings seem to be on the rise in many countries. Moreover, evolutional perspectives can help us to unravel this complex problem.

2.       The four sections outlined toward the end of the Introduction are all well and good but ambitious. It would be great if the author can find a way to link the four sections to create a sense of cohesiveness so that we are not just looking at 4 separate items. For example, how does evolutionary perspectives of the brain as apps help us to come up with the best therapeutical measures? It should not take too much work as the author has been working in this area for some time. But the effort will be well worth it.

3.       In section 2.2, Why We Have Feelings, the author describes, in essence, the co-evolution of emotion, feelings, and consciousness. What is left un-mentioned is the possibility of a whole host of other higher mental functions that may also be co-evolving. For example, it has been speculated but is difficult to prove, are consciousness, the need and the ability to express feelings through music and writing, and certain mental disorders apparently may be also part of this great co-evolution.

4.       On page 4, line 175, the author wrote, “ … negative feelings are pathological only if inappropriate or in excess.” The same should apply to positive feelings as well. Feelings are just feelings. Happy or sad feelings are the result of coloring by humans. As to how feelings become pathological, that coloring is only one of the parameters. In essence, obsession with any feeling, regardless of its color, can be problematic. Ask any parent how he or she would feel if their children should become a monk or nun, devoting their entire life to meditation. However, Gregor Mendel is always my favorite monk as well as a giant in the field of evolution.

5.       Figure 2, first appears on page 5, will appear again in different forms later. It may be useful for the author to tell the reader the implication of the color as well as the size of the bubbles right here. It may be even more productive if the figure can make some reference to or be somehow superimposed onto brain structure or systems, such as the limbic system, the neocortical cognitive system, etc. Then I was reminded that this implementation, although helpful at one level, can have problems at another. For example, I am struck by this thought as I was suddenly taken back to the days when lobotomy was in fashion. Lobotomy will reduce the size and eliminate the color of the bubbles, right?

6.       On page 5, line 215, the author wrote ‘This is presumably because it is in the interest of the genes to reside in a body/mind with a positive attitude to life.” I would argue that this is not correct. Genes only care about their survival (in terms of their molecular integrity) and propagation. In that sense, genes of the anxiety complex will behave the same way as genes of the HIV pathogen complex. At the level of genes, you might say that genes do not care about your state of mind, they don’t even care if you live for a long time or just a short amount time. They definitely do not care about the quality of the life of their host. They only care that you live long enough to help to propagate the genes.

7.       On page 8, the author mentions mismatches and discords (line 340). I would encourage the author specifically mention mental neglect and abuse, specially during development, as a major contributor of this mismatches and discords for the brain. For an organ such as the lung, smoking will be a major contributor of mismatch and discord. For the brain, nothing beats abuse during development.

8.       The contents on page 8 are great. The pattern of evolution of our society is complex. This evolution covers the style of parenting, how individuals deal with social life, and the very definition of social life. That said, it is not difficult to expect that all of the above will have some impact on the co-evolution of human traits at the genetic level, which impacts the prevalence of mental health conditions. One word of caution is that the term “evolution” in societal evolution and animal evolution or human evolution can have very different meanings. For example, one may be ultimately linked and explained at the genetic level. The others are more difficult to relate. This may be a particular important notion as the word “evolutionary” is the first word of the title of this review.

9.       The section on therapeutic measures is potentially the weakest link of the whole paper. First, it is the nature of the beast – this is a difficult part of the paper. Second, the writing looks tired. The paragraph starting on line 501 on page 11 is a fine example. “Focused exercise is needed ……..”  This is likely to be incorrect, particularly when the author has just made a good case on meditation in a convincing manner. Clearly, meditation should be classified as a blunt instrument, its action is unlikely to be precise or focused as the action of a scalpel. There is almost no information conveyed in the following sentence “A psychologist may help ……..”  Ditto for the next sentence “We live in a world ……” And so on. I suggest some tightening up work toward the end so the paper can finish in a high note.

10.   On Conclusion and Future Research, it is still unclear whether some of the genes influencing mental health conditions got to where they are today. Do they convey some yet unknown advantages to the human condition? While it is speculative that this may be the case in some of the genes involved in autism spectrum disorders, it is difficult to see that in those in the Down syndrome.

11.   A major contribution of the people who studied astronomy in the middle ages is that they encourage people to not think of the earth as the center of the universe. In evolution, the author mentions Stone Age frequently and perhaps too frequently. I think it is also helpful to also take the perspective of evolution from the point of the propagation of HIV. Thus, it is fine to consider the evolutionary perspective on mental health from the human side. It may also be productive to consider the same evolutionary phenomenon from the perspective of the genes themselves. After all, the human brain is just a host for the propagation of these genes. Right?

Author Response

2. The comment has been kept in mind while doing a general revision of the manuscript, hopefully improving the text in the desired direction.

3. I agree, but this is beyond the scope of the present text.

4. I agree. I have commented on cases where pos feelings can be considered pathological elsewhere in the paper, but more often people do not mind excessive happiness.

5. The Figure text has been expanded in that the use of the size of circles is indicated (the color is just added to make them stand out). Referencing them to brain anatomy would require a lot of text to be meaningful.

6. I certainly agree with the reviewer that genes do not care whether you are happy or sad. However, there can still be selection in favor of a positive default mood if this sentiment is more conducive to engaging in the tasks required for survival. The text has been clarified.

7. I agree with the reviewer, but I am cautious to expand the text by adding more examples.

8. I struggle with the expanded use of the term evolution myself. Unless otherwise stated, I use it for biological evolution - not, e.g., cultural evolution.

9. I have tried to rewrite this section. The reviewer may be right about the author being tired. It is also an enormous topic that in the present setting needs to be dealt with briefly.

10. The remark is well made, but the evolution leading to our various weaknesses is beyond the present scope. 

11. Yes, I agree that the human body, brain included, can be seen as a tool designed to propagate genes. I did once do quite a bit of research on the evolution of HIV - before I became a happiness biologist.

Reviewer 2 Report

Line 9 : Overall a great evolutionary perspective and a comprehensive survey of existing body of knowledge and practices. However, a "strategy" requires more than observations. A formulation of individual intervention strategies would be ideal.

52-53: what is 'reasonable' and how does the observation TRANSLATES that the functioning is not a contributor anymore and main culprit is well-being. Is there a comparative study with samples that do not support functioning of the disabled?

57 - define terms here (currently at line 205)

59 - Contradictory to line 52-53

63 - digression from line 52-53 claim that functioning is supported well

104 - please provide reference to the primary purpose of nervous system

129 - How can it be characterized as 'most' functions? Lines 87-90 suggest it may be complicated by multitude of nerve to app interactions

159 - unclear statement, please rephrase

205 - terms used before defining (line 57)

268 - does it makes model complex to apply in PREACTICE?

302 - also highlights complexity of conscious/nonconscious interactions

343 - a valid and important point

385 - while Stone Age examples are mostly valid to drive home the case in point, it tends to oversimplify and ignore implications of the modernization and evolution of human psychology. Same era rural joint family life to urban single family would make for a more relevant comparative.

365 - misplaced '?'

439 - please consider changing to 'since the Stone Age'

508 - how does the comprehensive strategy using these interventions differ from prevalent mental health practice

521 - is there a clinical study reference for the placebo claim?

605 - sudden change in voice - kindly rephrase

611 - not necessarily

Author Response

Line 9 is changed to read "I discuss an approach that may prove useful for ..."

52-53: The sentence has been rephrased

57: The defining terms are moved

59: I fail to see any contradiction

63: An explanatory sentence is added.

104: This is in my mind considered textbook information, there is no obvious suitable reference. I have used similar statements many times before without added reference.

129: The words "most if not all" are deleted

159: The sentence has been rephrased

205: Defining terms have been moved

268: I am uncertain as to the content of this comment

302: I agree with the reviewer's comment

343: Thank you

365: Corrected

439: Added "as well" instead

508: Have added "present" to clarify that it is about prevalent practice

521: Yes, the reference is at the end of the sentence

605: Sentence has been rephrased

611: Agree, have added "most"

Round 2

Reviewer 1 Report

Congratulations for your work. The manuscript is well done. Good luck to you and your readers.

Author Response

Thank you

Reviewer 2 Report

512 (was 508): "typical" is added not present (Have added "present" to clarify that it is about prevalent practice). I still feel a better connection and application to prevalent practice should be discussed. What changes to the prevalent practices are needed and being proposed?

Author Response

I have expanded the paragraph by pointing out the following in the hope of clarifying the issue. My aim is not to suggest concrete changes in practice but rather to offer a theoretical platform for understanding therapy.

Round 3

Reviewer 2 Report

Thanks you for clarifying and your patience through the review process.